# Frozen Ready-to-(h)eat Meals: Evolution of Their Quality during a Real-Time Short Shelf Life

**DOI:** 10.3390/foods12051087

**Published:** 2023-03-03

**Authors:** Ilenia Dottori, Stefania Urbani, Beatrice Sordini, Maurizio Servili, Roberto Selvaggini, Gianluca Veneziani, David Ranucci, Agnese Taticchi, Sonia Esposto

**Affiliations:** 1Department of Agricultural, Food and Environmental Sciences, University of Perugia, 06124 Perugia, Italy; 2Department of Veterinary Medicine, University of Perugia, 06124 Perugia, Italy

**Keywords:** ready meals, frozen, short shelf life, evolution of quality, texture, peroxide value, phenolic compounds, carotenoids, volatile compounds, sensory analysis

## Abstract

The purpose of this experimentation was to study the evolution of the quality of two types of blast-frozen ready-to-(h)eat meals, tortellini and a vegetable soup, during a short shelf life of 70 days. The analyses, performed in order to identify any variations resulting either from the freezing process or from the subsequent storage, carried out at the temperatures of −30 °C and −18 °C, respectively, examined the consistency of the tortellini and the soup, the acidity and the peroxide value of the oil extracted from them, the phenols and carotenoids present in the soup, the volatile compounds in the tortellini and the soup, and a sensory analysis of both products. The results showed that, during the 70 days of shelf life, there was no variation in the texture of the tortellini, but there were changes in the consistency of the soup, which decreased as the days of storage went on. Furthermore, statistically significant increases (*p* < 0.05) in the acidity and in the peroxide value of the oil of the soup were observed during the storage period; however, no statistically significant difference (*p* > 0.05) in the peroxide value of the oil of the tortellini was found. Moreover, no quantitative changes were observed in the phenolic compounds and carotenoids in the soup or in the volatile substances of either product. Finally, the sensory analysis confirmed, together with the chemical data, that the blast-freezing process adopted was suitable to maintain the good quality of these fresh meals, even if some technical modifications (in particular, lower freezing temperatures) should be adopted to improve the final quality of the products.

## 1. Introduction

The term ‘convenience food’ refers to ready-to-eat foods, such as ready-to-eat vegetables and salads, ready-made soups and sauces, frozen dietetic dishes, pizza, meals for microwaves, etc. These preparations satisfy the user’s need to speed up preparation times and are designed and appreciated especially by those who work, by single people, by those with little experience in the kitchen or little time available, and by the elderly [1]. More precisely, convenience foods can be sorted into two distinct categories depending on their degree of processing: partially prepared foods (ready-to-cook) and ready-to-eat/ready-to-(h)eat foods. These convenience products are, therefore, foods for which a significant fraction of the preparation time, cooking skills and necessary applied energy is performed by the food industry or retailers, instead of by people at home. Regarding the preservation of these products, freezing is one of the preferred processes, because it keeps the qualitative characteristics of the food almost intact and prolongs its shelf life [2]. Freezing is an operation by which a food is brought to a temperature below its freezing point and a part of the water in it undergoes a change in state, forming ice crystals. Extreme cold retards the growth of microorganisms and slows down chemical changes that can affect the food quality and cause its spoilage [3]. The main thermal events during the freezing process are accompanied by a reduction in the heat content of the material to be frozen, which cools down to the temperature at which the nucleation phase begins; indeed, before the ice formation, it is necessary to form a nucleus on which the crystal can grow. Once the first crystal appears in the solution, there is a phase change from liquid to solid with further crystal growth; thus, there is an accretion phase [3]. The number of crystallization nuclei increases with the increase in the undercooling interval, and the latter is longer the higher the freezing speed is. When thawed, the food retains its texture and intracellular fluids [3]. Very frequently, frozen ready meals are heated in a microwave oven as an alternative to using an electric oven. With this technology, heat is generated throughout the material, leading to faster heating rates and shorter processing times compared to conventional heating. Furthermore, in order to maintain a high nutritional value and a high level of quality, the technology selected for packaging should also be carefully considered and studied. Packaging technology is essential not only for food protection, but also for consumer convenience; indeed, the fusion of packaging technology with advanced materials and functional packaging has provided more convenience to consumers, and several innovative packaging materials have been introduced in the ready meal market, such as microwave susceptors, self-heating packaging, active and intelligent packaging, and biodegradable packaging [4]. It is evident that, especially in recent years, ready-to-(h)eat meals are becoming more common and the food industry is looking for technological solutions that guarantee both food safety and the high quality of these products. Therefore, the purpose of this experimentation was to investigate the evolution of the quality of two different types of frozen ready-to-(h)eat meals after microwave heating and during a real-time short shelf life lasting 70 days. Notably, the two products were very different from each other, both in their ingredients and in their nutritional composition; the tested samples were tortellini stuffed with meat with a cream and cooked ham sauce, and a creamy pumpkin and carrot soup made with barley. The quality of ready meals must be maintained not only while they are frozen and in the market, but also throughout their DMD. In this work, it was not possible to take that longer period into consideration, but a series of two-stage, real-time short shelf life experiments were performed. Such experiments are very useful for these particular products, since the company that supplied the two frozen ready meals works mainly with the HO.RE.CA channel, which generally uses large quantities of these products within short periods of time.

## 2. Materials and Methods

### 2.1. Products

The products used for this work were kindly donated by Pronto Green S.p.A., a company in Perugia (Umbria, Italy) that produces and markets frozen foods. For the purpose of carrying out this study, the following two products were used: fresh and frozen samples of pumpkin and carrot soup (325 g each), and fresh and frozen samples of tortellini with cream and cooked ham (300 g each). The ingredients of the soup were as follows: water, pumpkin 32%, carrots 12%, mashed potatoes, barley 4%, leek, sunflower seed oil and salt. Regarding the tortellini, the ingredients were as follows: fresh stuffed egg pasta cooked (55%) (ingredients: fresh egg pasta (50%) (ingredients: durum wheat semolina, egg (20%), wheat gluten), meat filling (32%) (ingredients: beef (27%), pork (20%), mortadella (18%), cheese, breadcrumbs, onion, celery, carrots, egg, sunflower oil, nutmeg, garlic, pepper), water and salt), cream (23%), rehydrated skimmed milk, cooked ham (3.6%), wheat flour, butter, cheese, salt and nutmeg. The two ready-to-(h)eat meals were taken to the Department of Agricultural, Food and Environmental Sciences of the University of Perugia, where the fresh samples were placed inside a cold room at the temperature of 4 °C, while the frozen samples were distributed inside two chest freezers at the temperature of −18 °C.

### 2.2. Real-Time Short Shelf Life Test

The analyses of the samples were carried out with the aim of performing a real-time short shelf life test, in order to determine whether there were any changes in the rheological and texture properties, and the sensory characteristics of the two products during their storage during the short-term period. The products were then employed as follows for analysis: at time 0, both the fresh samples and samples that had just undergone the freezing treatment were used; at time 1, frozen samples kept at a temperature of −18 °C for 35 days were used; at time 2, frozen samples kept at a temperature of −18 °C for 70 days were used. To carry out the rheological and texture analyses, the chemical analyses and the sensory analysis, the samples were prepared following the instructions given on the packaging. In particular, the tortellini with cream and ham were heated, still frozen, in a microwave oven, without opening the package, for 4 min at “high” power. The creamy pumpkin and carrot soup with barley was heated, still frozen, in a microwave oven, slightly lifting the protective film on the package, for 8 min at “medium-high” power. Both products were then mixed before being used for the analyses.

### 2.3. Texture Profile Analysis of the Tortellini

The ready-to-(h)eat tortellini were subjected to TPA analysis using the TVT 6700 Texture Analyzer machine (Perten, Stockholm, Sweden) consisting of a 672040 Compression Plate 40 mm stainless steel piston that exerts mechanical pressure on the samples. The instrument settings for the analysis were as follows: the sample height was 18.0 mm, the starting distance from the sample was 3.0 mm, the number of cycles was 2, the compression was 50%, the custom force or distance was 0.0 mm, the hold time was 2.0 s, the distance above the trigger was 1.0 mm, the initial speed was 1.0 mm/s, the test speed was 1.0 mm/s, the retract speed was 1.0 mm/s, the trigger force was 5.0 g, the data rate was 200.0 pps and the diameter was 0.0 mm. For the analysis, both the fresh tortellini and the frozen tortellini at time 0, time 1 and time 2 were previously heated in a microwave oven and allowed to cool to room temperature. Furthermore, only completely intact tortellini were taken into consideration and an attempt was made to eliminate the small pieces of ham which formed part of the sauce. During the TPA analysis, two forces were exerted on the tortellini, which gave the results “Peak Force A” (g) and “Peak Force B” (g), which represented the work necessary to deform the samples in the first and second compressions, respectively. The parameters obtained from the analysis were hardness, i.e., the maximum load detected during the first compression cycle; cohesiveness, which was the ratio between the work performed in the second compression cycle and the work performed during the first compression cycle, calculated by dividing the area under the second peak by the area under the first peak; springiness, i.e., the return of the compressed sample in the time between the end of the first compression cycle and the beginning of the second cycle, measured as the compression distance to arrive at “Peak Force B” divided by the compression distance to arrive at “Peak Force A”; adhesiveness, which was the work required to overcome the attractive forces between the sample and the piston surface during the return of the first cycle; gumminess, which was the result of hardness x cohesiveness; and chewiness, which was the result of gumminess x springiness [5].

### 2.4. Evaluation of the Consistency of the Soup

The evaluation of the consistency of the soup was determined using a Bostwick viscometer consistometer (Greensenselab, Wagram an der Donau, Austria), an instrument equipped with a 2 mm thick stainless steel tank and a capacity of 100 mL, with adjustment feet and a level bubble supplied. A movable bulkhead separated the sample space from the 23 cm long slide lane, graduated in 0.5 cm fractions. Each sample, after being heated in a microwave oven (as described in Section 2.2) and allowed to cool down to room temperature (about 20 °C), was placed on the tray closed by the bulkhead, taking care to even out the level with a spatula. After releasing the bulkhead by acting on the snap system, the sliding stroke of the sample was read over a time of 30 s and the operation was repeated twice for each sample.

### 2.5. Determination of the Acidity and the Peroxide Value of the Oil Extracted from Soup and Tortellini

For the extraction of the oil from the soup and tortellini, 100 g and 70 g of product were taken, respectively, and mixed with hexane in a 1:3 ratio (300 mL for the soup and 210 mL for the tortellini), using ULTRA-TURRAX T25-IKA for 1 min at 17,000 rpm, and subsequently kept under stirring at 215 rpm at room temperature for 30 min. The mixture thus obtained was filtered with Perfect 2 Cordenons filter paper (Milan, Italy) and the filtrate was evaporated under a vacuum at 35 °C until the complete evaporation of the solvent. The acidity and the peroxide value were determined for the residual oil in accordance with the provisions of EU Reg. 2019/1604 [6].

### 2.6. Extraction and Evaluation of Phenols and Carotenoids in the Soup

The extraction of the phenolic compounds and carotenoids present in the soup was carried out, as reported in the work of Motilva et al. [7], making the following modifications: 5 g of soup was homogenized with ULTRA-TURRAX T25 for 1 min at 17,000 rpm with 10 mL of a hexane/acetone/ethanol mixture (50:25:25 *v*/*v*). The homogenate was centrifuged at 9000 rpm for 10 min, then the supernatant was placed in a separatory funnel and 10 mL of water was added, and the two phases were separated and further purified, as reported by Motilva et al. [7]; the supernatants obtained from each phase (organic and aqueous) were evaporated by rotavapor. The organic extract obtained was used for the determination of carotenoids, while the aqueous extract was used for the evaluation of phenolic compounds. The determination of carotenoids was carried out using an Agilent Technologies HPLC instrument model 1100, consisting of a quaternary pump complete with degasser, autosampler, thermostatted column compartment, UV-Vis (DAD) and fluorescence (FLD) photodiode detector. A Lichrospher Si 60 250 × 4 mm normal-phase column with a particle diameter of 5 μm (Merk KgaA, Darmstadt, Germany) was used for the analysis of β-carotene. Before being analyzed in HPLC, the sample was filtered with 0.22 µm PVDF syringe filters (Carlo Erba Reagents, Milan, Italy); the injected volume was 50 μL and the eluent flow was 1.3 L/min, using a mixture n-hexane/isopropyl alcohol (99.5:0.5 *v*/*v*) (solvent A) and n-hexane/isopropyl alcohol (70:30 *v*/*v*) (solvent B). The gradient was varied as follows: 100% solvent A and 0% solvent B for 2 min, in 8 min, 95% solvent A and 5% solvent B, in 5 min, 25% solvent A and 75% solvent B for 4 min and back to the initial conditions in 3 min, maintained for 5 min. The total chromatographic run time was 35 min. The ChemStation, also from Agilent Technologies, in addition to controlling the entire instrumentation, performed the processing of the chromatographic data (Agilent Technologies, Palo Alto, CA, USA). For the determination of β-carotene and lutein, the DAD was set at 450 nm. The quantification of β-carotene and lutein was performed using the calibration curves of the standard compounds, and the results are expressed as mg/kg. The evaluation of the phenolic compounds that were present in the aqueous extract obtained from the soup was carried out with the same HPLC instrumentation reported previously for the determination of β-carotene and lutein. The aqueous extract was filtered with 0.22 µm PVDF syringe filters (Carlo Erba Reagents, Milan, Italy), and the analysis and determination of the phenolic compounds was performed as reported by Taticchi et al. [8].

### 2.7. Evaluation of Volatile Compounds in Soup and Tortellini

The evaluation of the composition of volatile substances in the soup and the tortellini was carried out by mass spectrometry analysis coupled to gas chromatography (GC/MS) through headspace sampling by solid phase microextraction (SPME). The samples (2 g) were placed in 20 mL vials, 1 mL of a saturated solution of NaCl and 100 μL of an internal standard (4-methyl-2-pentanol 750 μg/L) were added, and the vials were hermetically closed and placed in the autosampler. SPME sampling of volatile compounds was performed by exposing the fiber consisting of Carboxen/divinylbenzene/polydimethylsiloxane 50/30 μm, 2 cm long (Supelco Inc., Bellefonte, PA, USA). Before adsorption, the sample was kept under stirring (750 rpm) for 30 min at 40 °C. The adsorbed compounds were then thermally desorbed for 5 min by inserting the fiber into the GC injector maintained at 250 °C. The analyses were carried out with an Agilent Technologies 7890B GC, equipped with a “Multimode Injector” (MMI) 7693A (Agilent Technologies, Santa Clara, CA, USA) and a thermostatted PAL3 RSI 120 autosampler equipped with a fiber conditioning module and shaker (CTC Analytics AG, Zwingen, Switzerland); this was coupled to a single quadrupole MS 5977B (MSD) with an XTR (Extractor Ion Source) electron impact source (Agilent Technologies, Santa Clara, CA, USA). For the separation of the volatile compounds, a DB-WAXetr fused silica capillary column was used, with a length of 50 m, an i.d. of 0.32 mm and a film thickness of 1 μm (Agilent Technologies, Santa Clara, CA, USA). Helium was used as carrier gas at a flow of 1.7 mL/min, which was kept constant throughout the analysis time by means of an Electronic Flow Control (EFC) device. The GC column oven temperature was set according to the following schedule: the initial temperature was 35 °C and maintained for 4 min, then increased to 45 °C at 5 °C/min, further increased to 150 °C at 4 °C/min, further increased up to 180 °C at 8 °C/min, maintained for 2 min, and finally brought to 210 °C at 11 °C/min and maintained for 13.77 min; under these conditions, the total analysis time was 55 min. The injector was set at a temperature of 250 °C. The temperature of the “transfer line” was set at 215 °C; regarding the experimental conditions of the mass spectrometer, the temperature of the source was 190 °C and that of the quadrupole was 150 °C. The electron impact mass spectrum (EI) was recorded with an ionization energy of 70 eV in the mass range 25–350 a.m.u., 4.3 scans/s, and the MS spectra were acquired in scan mode. The processing of the collected data was carried out using the Agilent MassHunter B.08.00 software with the Unknown Analysis module. The identification of volatile compounds was performed by comparing the mass spectra and retention times thus obtained with those of pure analytical standards and with the spectra of the NIST-2014 library. The volatile compounds were quantified and expressed in μg/kg by comparing the area of each peak of the extracted ion (corresponding to each compound evaluated) with the ion area of the peak of the internal standard (4-methyl-2-pentanol), as reported by Xiao et al. [9].

### 2.8. Sensory Analysis

At the Department of Agricultural, Food and Environmental Sciences of the University of Perugia, for both the tortellini and the soup, a triangle test according to the ISO 4120:2021 standard [10] was performed on the fresh and frozen samples at time 0 after being heated in a microwave oven (as described in Section 2.2) and cooled to room temperature; this was in order to observe whether 25 judges were able to distinguish the fresh product from the frozen one. Each panel member was presented with three samples that were coded differently, two of which were identical and one was different. In this type of test, the taster has the task of identifying the different samples; even if he is unable to do so, he must still give an answer (forced choice). The three samples were presented accompanied by a sheet, which provided instructions for carrying out the test and which indicated, according to the tasting sequence, the codes of the samples to be examined [11]. The criteria for determining the significance of the freezing treatment were based on tables for the binomial distribution of the triangle test. For this test, 25 tasters (12 men and 13 women aged between 25 and 55) were employed and the results were considered significant by applying a risk factor α equal to 0.01 with probability Pd 30% (percentage of correct answers beyond chance, upper limit value of the one-tailed confidence interval corresponding to β = 0.01). The results were elaborated through the tables of significance for this type of test. The form used by the tasters for this test is shown in Appendix A.

A quantitative descriptive sensory analysis (QDA) was also performed on both products when fresh, frozen at time 0, frozen at time 1 and frozen at time 2, according to the ISO 13299:2016 standard [12] and using 10 expert panelists (4 men and 6 women aged between 25 and 55) who had been trained to recognize and quantify the sensory characteristics of each of the two products. The number of trained panelists was decided based on the amount of product available. The evaluations were conducted in a classroom with single stations, one for each judge, where approximately 40 g of product was served for each sample after being heated in a microwave oven (as described in Section 2.2), cooled to room temperature, and then placed in plastic tasting glasses marked by an alphanumeric code. The quantitative descriptive sensory analysis was performed using a specific sensory analysis form for each product. The specific descriptors for appearance, odor, taste, texture, trigeminal sensations and final sensations were included on the two forms (Appendix A). The different attributes were quantified on a 9 cm unstructured ordinal intensity rating scale.

### 2.9. Statistical Analysis

To compare the results obtained in the experimentation and test the differences between the different products, the Tukey test was used; it was performed with SigmaStat v.2.0 software. PanelCheck software version 1.4.2 (Nofima, Trømso, Norway) was used for the statistical processing of sensory analysis data, which took place through the application of PCA (principal component analysis).

## 3. Results

### 3.1. Evolution of the Texture of the Tortellini

The data concerning the structural characteristics of the tortellini are reported in Table 1, but they have an important standard deviation; this depends on a whole series of extrinsic and intrinsic variables linked to the product itself and to the conditions in which it was found. Indeed, the product analyzed had a condiment, i.e., a sauce made from cream and cooked ham, which was distributed in a heterogeneous way on the tortellini. For this reason, the tortellini analyzed could have different amounts of sauce. Furthermore, the shape of the container used for their packaging and heating could result in not all the tortellini receiving the same amount of heat and not heating up in the same way. Another important factor was that tortellini are a type of filled pasta; therefore, the meat could occupy a space inside the tortellini that was not exactly the same for each of them. For these reasons, the matrix examined was very heterogeneous, and this could explain the high values obtained for the standard deviation. It can be observed how the hardness value for the sample at time 2 decreased by 31.4% compared to that for the fresh sample. As was also reported in the work of Zhang et al. [13], hardness is an important parameter of dough quality; the lower the hardness in a given range, the softer the dough. Therefore, increasing the hardness does not improve the dough [13]. It can be assumed that there was a decrease in hardness during the frozen storage of the product because some macrocrystals could be formed during the storage period that, at the time of thawing, probably damaged cell walls and thus had an effect on the consistency of the product. Regarding the springiness, chewiness and gumminess of the sample at time 2 compared to the fresh sample, these decreased by 32.8%, 56.4% and 34.4%, respectively. In particular, the decrease in chewiness may have happened because the gluten network was damaged during frozen storage, leading to the increase in the leaked soluble substance on the noodle surface when reheating [14].

### 3.2. Evolution of the Consistency of the Soup

Figure 1 shows that there was a statistically significant difference (*p* < 0.05) in the texture of the soup that had been frozen compared to the fresh soup. The distance traveled along the slide lane by the fresh soup was equal to 4.5 cm, while for the frozen one at time 0, the value was equal to 11.1 cm. A value comparable to the latter was also found at time 1 (11.0 cm), while there was a further increase in the distance traveled by the frozen soup at time 2, equal to 15.3 cm. This result was in accordance with what was reported in the work of Araújo-Rodrigues et al. [15], which examined baby carrot and cherry tomato pulps, studying the impact of the freezing process and monitoring parameters such as consistency over 6 months of storage. The results of that study showed that, in both cases, a significant decrease in viscosity was observed after freezing. In the carrot pulp, although the decrease in viscosity was more evident in the first months of freezing, there was a significant decrease throughout the storage period, except from the fifth to the sixth month of storage when there were no significant changes in viscosity values. Regarding the cherry tomato pulp, the variation in viscosity was even more evident in the first months of frozen storage, but the results suggested significant variations in all the months analyzed. This was possibly because the ice crystals produced during the freezing process could have caused damage to the integrity of the cell membranes, and because the consequent loss of water from the intracellular compartments affected the physicochemical and physical properties of the product, such as viscosity. In general, lower material properties are found in frozen vegetable foods than in their fresh equivalents [15].

### 3.3. Evolution of Acidity and Peroxide Values of the Oil Extracted from Soup and Tortellini

Figure 2a shows a statistically significant progressive increase (*p* < 0.05) in the acidity (%) of the soup over time, from a value equal to 0.71% for the fresh soup to 1.40% for the frozen soup tested at time 2 (+97.2%). This could be because the product was processed using a freezing technology, but the temperature at which it was stored was −30 °C in a forced air cell for an entire night. This means that there was the formation of a smaller number of crystals compared to an ultra-rapid freezing process (which can employ temperatures of −50 °C/−70 °C) and that these crystals were also larger in size. It is plausible that, as the storage time of the product at a temperature of −18 °C increases, these crystals may grow further; at the time of thawing, there may then be a greater rupture of the cell membranes, with the consequent release of the cellular content, in which there are intracellular enzymes such as lipases. The lipases act on triglycerides by hydrolyzing the ester bond and leading to the formation of free fatty acids, which increase the acidity of the products. In this case, the lipases could not be completely inactivated because the freezing occurred at temperatures that were not low enough. Therefore, since the freezing process lasted overnight, there could be some areas of free super-concentrated water that was not completely crystallized and was available for enzymes.

Regarding the peroxide values, a parameter evaluated to measure the degree of the oxidative rancidity of fats, Figure 2b shows that there was a statistically significant (*p* < 0.05) increase when comparing both the fresh and frozen soups at time 0 with the frozen soups at times 1 and 2 of storage. Thereafter, the value remained unchanged from time 1 to time 2 of storage.

In Figure 3, when comparing the fresh product to the frozen product at storage time 2, it can be observed that the peroxide value in the tortellini had increased by 16.7%, but the data were not statistically significant (*p* > 0.05). Data relating to the peroxide values suggested that the products were well protected from oxidation by oxygen. This could depend mainly on the packaging used, which included a film that sealed the container and ensured that the products were not only protected from oxygen, but also from other factors that could promote the oxidation of fatty acids, such as light. There was a statistically significant increase (*p* < 0.05) in the number of peroxides in the soup, but not in the tortellini because of the different types of fatty acids contained in the two products. The tortellini contained mainly fats of animal origin (derived from the meat and cream used), so they were mostly saturated fatty acids, which are more stable with respect to oxidation. In the soup, on the other hand, the ingredients were all of vegetable origin, so there was a greater content of mono and polyunsaturated fatty acids, which are more unstable and more easily subject to oxidation.

### 3.4. Evolution of Phenolic Compounds and Carotenoids in the Soup

Regarding the content of phenolic compounds and carotenoids in the soup, the molecules extracted and evaluated from the product were β-carotene, lutein, α-tocopherol, quercetin-3-o-rutinoside and quercetin. Table 2 shows that there seemed to be a progressive tendency towards a decrease in these compounds over time; comparing the fresh product with the frozen at time 2, the β-carotene underwent a decrease of 3.1%, lutein decreased by 4.2%, α-tocopherol decreased by 3.0%, quercetin-3-o-rutinoside decreased by 1.7% and quercetin decreased by 11.1%. However, the decreases observed were not statistically significant (*p* > 0.05); therefore, it could be assumed that the freezing process did not have a negative impact on the nutritional and health qualities linked to the carotenoid and phenol contents of the soup, and especially that there were no statistically significant differences even during the storage period of 70 days at −18 °C. The work of Im et al. [16] analyzed the total phenol content of frozen potatoes and carrots treated with different pre-treatment and thawing methods. From the results of that study, it was seen that there was no significant difference in the total phenol content of potatoes after thawing (*p* > 0.05); however, when taking into consideration different pre-treatments, the phenol content was between 2.07 and 2.09 µg GAE/mg in the control (not pre-treated before freezing), between 1.44 and 1.54 in the HB product (blanched in hot water at 100 °C), and between 1.59 and 1.72 in the SB product (blanched in steam at 100 °C). During the heat treatment, the total phenol content showed a tendency to decrease compared to the control. One study has shown that when a lotus root was blanched in hot water, the total phenol content decreased with increasing the heat treatment time, and in another work, it was evident that blanching within 5 min could preserve the nutritional content. The difference in the total phenol content between the same pre-treatment groups was 0.02~0.13 µg GAE/mg, and no significant change was observed according to the thawing method. Therefore, it was concluded that the total phenol content of potatoes after thawing varied depending on the pre-treatment method, rather than on the thawing method. Regarding the polyphenol content of carrots, this did not show a clear trend according to the pre-treatment method or the thawing method; however, when pre-treated with SB and thawed in running water, they did show a higher total phenol content, at 1.45 ± 0.21 µg GAE/mg [16].

### 3.5. Evolution of Volatile Compounds in Soup and Tortellini

Figure 4 shows that the content of volatile substances in the fresh soup was 89.7% terpenes, followed by 4.4% aldehydes, 2.5% furans, 1.8% alcohols, 0.9% sulfur compounds and 0.7% ketones. The total of volatile compounds in the fresh soup was equal to 14,852.5 µg/kg (expressed as 4-methyl-2-pentanol).

Table 3 highlights the fact that most of the volatile substances belonging to the various classes of compounds remained unchanged during the real-time short shelf life test, and that there were no statistically significant differences (*p* > 0.05) between the fresh soup and the frozen soup at storage time 2. Observing the data, however, it is possible to see how some substances gradually increased; for example, hexanal, which is one of the main oxidation indices for food products, had increased by 11.0%. The same trend was also found for other aldehydes linked to oxidation, such as pentanal (+0.4%), nonanal (+2.6%) and 2,4-decadienal (+21.8%), while there was a small decrease in (E)-2-heptenal (–1.2%). In general, there was an increase of 6.3% in the total aldehydes between the fresh product and the frozen product tested at storage time 2, but the data were not statistically significant (*p* > 0.05). The alcohols in the soup, in particular 2-methyl-1-butanol, 1-pentanol and 1-hexanol, were probably derived from the barley grains it contained. Indeed, as reported in the study by Cramer et al. [17], alcohols are the major quantitative constituents of barley volatiles, followed by aldehydes, ketones and furans. The alcohol content in the soup appeared to have a decreasing trend over time (–4.7% in the frozen soup at time 2 compared to the fresh soup), but the data were not statistically significant (*p* > 0.05). Regarding ketones, in the soup there was only acetoin, which decreased by 4.7% at storage time 2; however, in that case, the result was also not statistically significant (*p* > 0.05). The most abundant compounds in the soup were terpenes, such as limonene, α-pinene, β-pinene, β-myrcene, γ-terpinene and caryophyllene. This was because the product was completely plant-based and one of the main ingredients was carrots. As reported in the study by Kjeldsen et al. [18], which examined changes in the volatile compounds of carrots (*Daucus carota* L.) during refrigerated and frozen storage, terpenoids, such as monoterpenes, sesquiterpenes and irregular terpenes, were the predominant class of volatile compounds in carrots in terms of number and quantity. Terpenoids accounted for more than 99% of the total volatile mass in carrots. The main monoterpenes were α-pinene, sabinene, β-myrcene, limonene, γ-terpinene, p-cymene and terpinolene [18]. These results agreed fairly well with the data from the present study. In general, all plants are naturally rich in terpenes, as the latter perform important biological activities. Indeed, these metabolites have an important role in nature since they can act as an indirect form of defense in plants against herbivores and pathogenic microorganisms, but also against extreme abiotic factors such as temperature, sun exposure, humidity and a lack of nutrients [19]. The total number of terpenes seemed to undergo a very small decrease of 0.2%, but the data were not statistically significant (*p* > 0.05). Inside the fresh product there was also a small amount of furans (2.5%), which probably derived from the cooking process to which both the fresh and the frozen soup were subjected. The quantity of furans underwent an increase of 16.7% during the short-term shelf life up to time 2, but this result also was not statistically significant (*p* > 0.05). Another small percentage of volatile substances found in the soup product was indicative of sulfur compounds (0.9% in fresh soup), in particular dimethyl sulfide and dipropyl disulfide. Generally, sulfur compounds are found as secondary metabolites in both plants and microorganisms, where they mostly have pleasant odors rather than offensive ones. These compounds are distributed in some plant families, including *Brassicaceae*, *Apiaceae*, *Liliaceae*, *Caricaceae*, *Capparaceae*, *Solanaceae* and *Rutaceae* [20]; therefore, it is not surprising to find small quantities of them in plant-based products, such as the soup under examination in this work. The amount of sulfur compounds had an increasing trend (+16.2% at storage time 2), but the data were not statistically significant (*p* > 0.05). In general, when observing all the volatile compounds present in the soup at storage time 2, there were no statistically significant changes in them (*p* > 0.05) compared to the fresh product.

As shown in Figure 5, even for the fresh tortellini, the content of volatile substances was mostly made up of terpenes (92.2%), followed by ketones (5.6%), aldehydes (1.1%), alcohols (0.6%) and organic acids (0.5%). The total amount of volatile compounds in fresh tortellini was 31,556.0 µg/kg (expressed as 4-methyl-2-pentanol).

As was the case with the soup, in the tortellini (Table 4), there was also an increase in the total number of aldehyde compounds (+9.5%) when comparing the fresh product to the frozen one at storage time 2, but the data were not statistically significant (*p* > 0.05). In particular, there was an increase in aldehydes linked to oxidative processes, such as pentanal (+12.0%), hexanal (+10.2%), (E)-2-heptenal (+41.3%), nonanal (+16.1%) and 2,4-decadienal (+27.2%); however, in this case, the results were also not statistically significant (*p* > 0.05). Moreover, in the tortellini, there was a small percentage of alcohols (0.6% in the fresh product), in particular 1-pentanol and 1-hexanol, since the wheat flour used for the production of the tortellini contained volatile compounds such as alcohols and ketones [17]. The alcohol content of the tortellini showed a decreasing trend (–5.3%) during the storage period up to time 2, but the data were not statistically significant (*p* > 0.05). Regarding ketones, these constituted a greater percentage of the total volatile substances in the tortellini (5.6% in the fresh ones) than was found in the soup. This was not only because of the ketone content in the volatile compounds of the wheat flour used in the dough of the tortellini, but also because this dish had a sauce (cream and cooked ham) that contained a high quantity of lipids that could undergo oxidative processes. The ketones (acetoin, 2-pentanone, 2-heptanone, 2-nonanone) seemed to undergo an increase of 3.3% during storage up to time 2, but the data were not statistically significant (*p* > 0.05). In the tortellini, as in the soup, most of the volatile substances found were terpenes (92.2% in the fresh product); in this case, their presence was mainly linked to the spices present in the filling of the tortellini, as well as to those used to prepare the sauce used as a condiment. During the storage period, the total number of terpenes tended to decrease (–1.0%) compared to the fresh product, but the data were not statistically significant (*p* > 0.05). A very small percentage of volatile substances, equal to 0.5% in the fresh tortellini, was made up of organic acids, such as acetic acid, butanoic acid and pentanoic acid. The presence of these last two organic acids was mainly linked to the cream present in the sauce of the tortellini. There was an increase in the sum of organic acids at storage time 2 compared to the fresh product (+3.3%), but this result was not statistically significant (*p* > 0.05).

### 3.6. Sensory Analysis 

In order to determine any sensory differences related to the freezing treatment, the first sensory evaluation consisted of a discriminant triangle test. Table 5 shows that there was a significant difference between the fresh and frozen versions of both types of ready-to-(h)eat products studied. Of the 25 expert panelists who served as judges, 22 were able to identify the “different” sample when a factor β = 0.01 was applied, while 17 was the maximum number of correct choices to conclude that there was a similarity between the samples, according to ISO 4120:2021 [10]. 

The next step was to understand in what ways and how much the products differed. For this reason, further sensory evaluation consisted of a quantitative descriptive sensory analysis performed by a trained sensory panel (10 panelists), which compared all the samples of soup and tortellini in terms of their appearance, odor, taste and texture/mouthfeel attributes. The collected data were statistically analyzed using principal component analysis (PCA). Two biplots illustrate the results of the PCA: the first one is for the visualization of the soup samples analyzed (Figure 6) and the second one is for the tortellini (Figure 7).

Figure 6 shows that the first two principal components (PC1 and PC2) explained 88.6% of the total variability. In addition, it is possible to observe how the samples were distributed differently depending on whether or not they were subjected to the deep-freezing treatment, and also depending on the storage time. In particular, along the first principal component (PC1), there was a clear distinction among the fresh soup sample (NFS), the frozen soup samples at time 0 (FS T0) and at time 1 (FS T1), and the frozen sample at time 2 (FS T2). Furthermore, along the second principal component (PC2), there was a clear distinction, especially for the FS T0 sample; this was located in the lower part of the biplot compared to the NFS, FS T1 and FS T2 samples, which were located in the upper part. The variables that had the greatest weight in differentiating the NFS sample along the first component (PC1) were adhesiveness, creaminess, viscosity, softness and consistency. The FS T0 sample, on the other hand, differed in variables such as moisture and homogeneity of color, while the FS T1 sample was also distinguished by variables such as consistency, chewiness, crunchiness (of the barley grains contained in it), fresh vegetable odor, potato odor and cohesiveness. Finally, in the FS T2 sample, all the descriptors that characterized the NFS, FS T0 and FS T1 samples were less relevant; in particular, this sample differed visually, due to its translucency/wateriness. Along the second component (PC2), the differentiation of the samples was reconfirmed based on the storage time in the freezer at a temperature of −18 °C.

Figure 7 shows that the first two principal components (PC1 and PC2) could help explain 82.1% of the total variability in the tortellini samples. Furthermore, it is possible to observe how, in this case also, the samples were distributed in a different way depending on whether or not they were subjected to the deep-freezing treatment and also as a function of the conservation time. In particular, the samples frozen at time 1 (FT T1) and at time 2 (FT T2) were on the right side of the first principal component (PC1), opposite to the sample frozen at time 0 (FT T0) and the fresh sample (NFT), which were located in the left part of the first component (PC1); similarly, they were found distributed in the upper and the lower parts of the second principal component (PC2), respectively. The short shelf life for tortellini allowed us to identify, along the first principal component (PC1), a similarity between the fresh sample and the freshly frozen sample; this is the case even if after a certain storage time, in the FT T1 and FT T2 samples, it was possible to notice some differences related to the concentration of certain descriptors, including aftertaste, persistence, umami, meat odor, vegetable odor and moisture. The FT T0 sample differed from the others for variables such as the bright yellow color of the pasta, firmness and the smoothness/sliminess of texture. Finally, the NFT sample was located in the lower left quadrant and differed along the first principal component (PC1) for visual characteristics, such as the clotted cream of the sauce and being overcooked; meanwhile, along the second principal component (PC2), variables related to texture, such as chewiness, springiness and gumminess, were also distinguished. It seems that the freezing process did not have a negative impact on tortellini even if it was not ultra-rapid, probably due to the lower water content in this type of food. Indeed, if water had been present in larger quantities, it could have been subject to migration and might have created more difficult areas to freeze.

## 4. Conclusions

To the best of our knowledge, this is the first scientific work that has fully evaluated the impact of the freezing process and subsequent microwave heating on the rheology, chemical composition and sensory quality of ready-to-(h)eat meals. From the data obtained in the experimentation, the influence that the freezing process can have on ready-to-(h)eat meals can be seen; in addition, and above all, how the quality of this type of product can evolve during a real-time short shelf life can also be observed. The rheological analyses of the tortellini highlighted that the fresh product was harder, more elastic, more chewable and gummier than the frozen ones, data that were corroborated by the results of the sensory analysis. As far as the consistency of the soup was concerned, in that case also the sensory analysis data reflected the results of the instrumental analyses; the fresh soup appeared to have a greater consistency than the frozen ones at all three storage times, and notably, the consistency seemed to decrease with the increase in the days of storage. The chemical analyses underlined that there was an increase in the acidity of the oil in the soup, probably linked to the temperatures and freezing times of the product. In both the soup and the tortellini there was also an increase in the peroxide values, particularly at storage time 2, but the data were statistically significant (*p* < 0.05) only in the case of the soup. Regarding the content of phenols and carotenoids in the soup, the decreases found over the days of storage were not statistically significant (*p* > 0.05). Finally, the variations in the volatile compounds of the two products were also not statistically significant (*p* > 0.05). The sensory analysis data confirmed those data obtained from chemical and rheological analyses, and in some cases, they added information that was not examined at an instrumental level. In conclusion, the freezing process and the subsequent storage time at −18 °C during a real-time short shelf life test lasting 70 days did not affect the general acceptability of the two products, but only had an impact on some sensory characteristics. In general, it should be emphasized that the final quality of the products strongly depended on the temperature and the duration of the freezing process; the lower the temperatures and the shorter the freezing times, the more the damage caused by freezing was reduced, as numerous scientific works have amply demonstrated. Since the products analyzed were products from a small–medium-sized company, it was normal that the systems used for freezing utilized a blast-freezing method that did not drop below −30 °C, unlike large industries that often have much more powerful equipment in terms of operating capacity and product quality. One useful technological improvement could be to circulate the air in the cold room at a higher speed so as to allow the process to have a shorter duration. Finally, it should also be noted that the products were already packaged when they underwent the freezing process, so the packaging barrier should also be taken into consideration.

## Figures and Tables

**Figure 1 foods-12-01087-f001:**
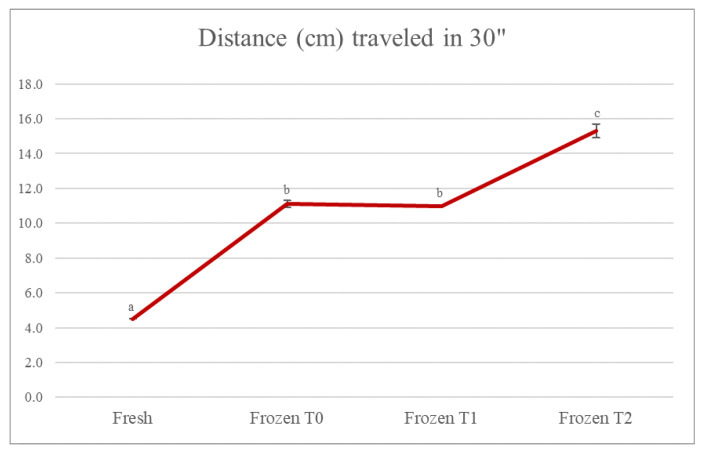
Evolution of the consistency of the soup. The results are the mean of two determinations ± the standard deviation. The comparison between the different storage times was carried out by one-way ANOVA. Different lowercase letters indicate a statistically significant difference (*p* < 0.05). Fresh = fresh soup (not frozen); Frozen T0 = frozen soup at the time of freezing treatment; Frozen T1 = frozen soup after 35 days of storage; Frozen T2 = frozen soup after 70 days of storage.

**Figure 2 foods-12-01087-f002:**
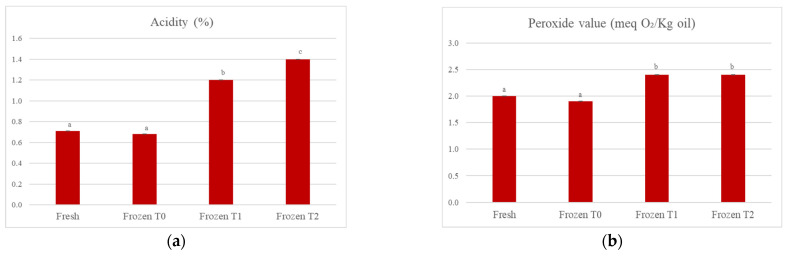
(**a**) Evolution of acidity of the oil extracted from the soup; (**b**) evolution of peroxide value of the oil extracted from the soup. The results are the mean of two determinations ± the standard deviation. The comparison between the different storage times was carried out by one-way ANOVA. Different lowercase letters indicate a statistically significant difference (*p* < 0.05). Fresh = fresh soup (not frozen); Frozen T0 = frozen soup at the time of freezing treatment; Frozen T1 = frozen soup after 35 days of storage; Frozen T2 = frozen soup after 70 days of storage.

**Figure 3 foods-12-01087-f003:**
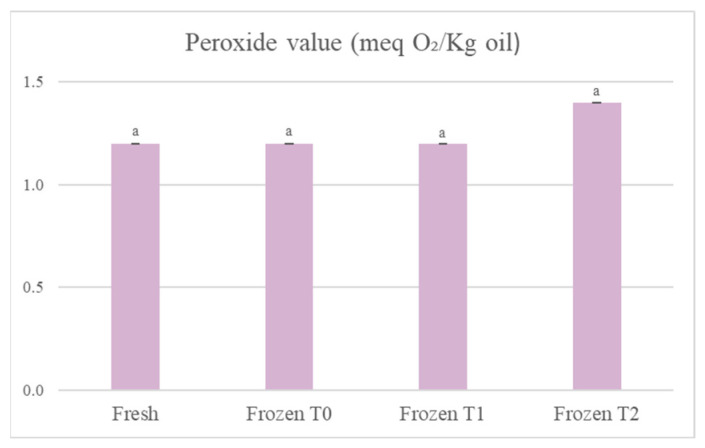
Evolution of the peroxide value of the oil extracted from the tortellini. The results are the mean of two determinations ± the standard deviation. The comparison between the different storage times was carried out by one-way ANOVA. Different lowercase letters indicate a statistically significant difference (*p* < 0.05). Fresh = fresh tortellini (not frozen); Frozen T0 = frozen tortellini at the time of freezing treatment; Frozen T1 = frozen tortellini after 35 days of storage; Frozen T2 = frozen tortellini after 70 days of storage.

**Figure 4 foods-12-01087-f004:**
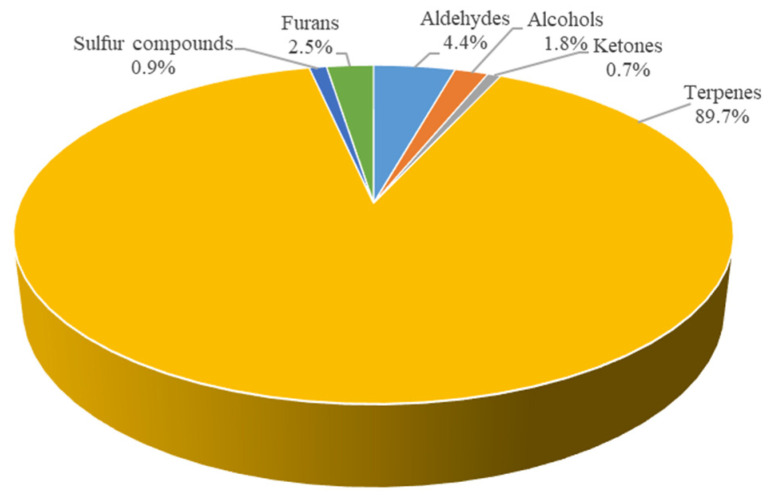
Percentage distribution of volatile compounds in the fresh soup.

**Figure 5 foods-12-01087-f005:**
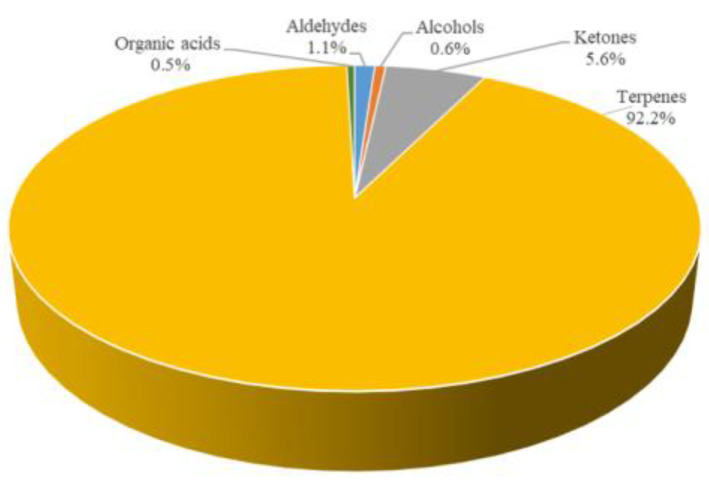
Percentage distribution of volatile compounds in the fresh tortellini.

**Figure 6 foods-12-01087-f006:**
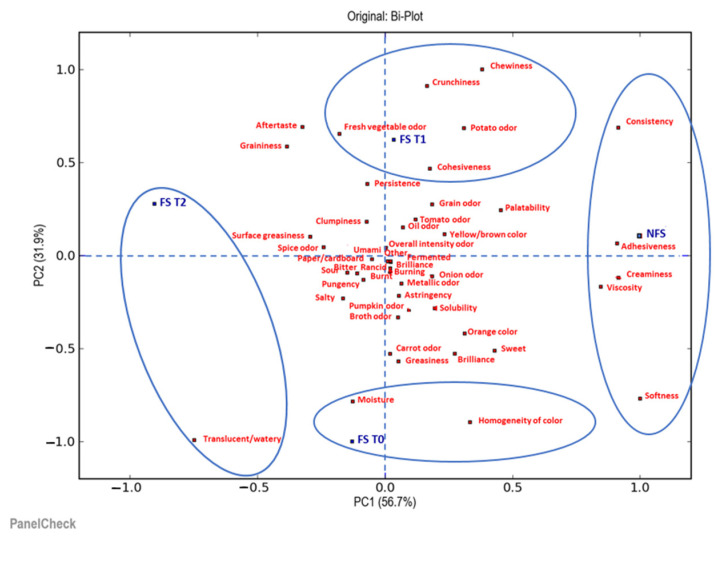
Biplot of the two most significant principal components, PC1 and PC2, by a principal component analysis (PCA) of the descriptive parameters of the soup samples analyzed. NFS = fresh soup (not frozen); FS T0 = frozen soup at the time of freezing treatment; FS T1 = frozen soup after 35 days of storage; FS T2 = frozen soup after 70 days of storage.

**Figure 7 foods-12-01087-f007:**
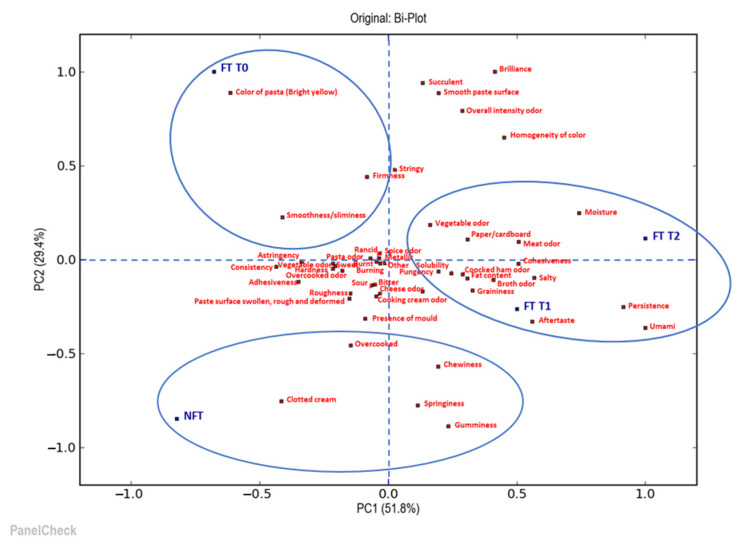
Biplot of the two most significant principal components, PC1 and PC2, by principal component analysis (PCA) of the descriptive parameters of the tortellini samples analyzed. NFT = fresh tortellini (not frozen); FT T0 = frozen tortellini at the time of freezing treatment; FT T1 = frozen tortellini after 35 days of storage; FT T2 = frozen tortellini after 70 days of storage.

**Table 1 foods-12-01087-t001:** Evolution of the texture of the tortellini *.

	Fresh	Frozen		
		T0	T1	T2
Peak Force A (g)	1000.3 ± 301.5 a	573.3 ± 231.2 b	611.5 ± 249.1 b	686.2 ± 254.0 b
Springiness	0.8 ± 0.1 a	0.5 ± 0.2 b	0.6 ± 0.1 ab	0.5 ± 0.1 b
Adhesiveness	0.3 ± 0.4 a	41.8 ± 34.7 b	4.6 ± 6.7 a	46.6 ± 29.8 b
Cohesiveness	0.5 ± 0.1 ab	0.5 ± 0.1 a	0.6 ± 0.1 b	0.5 ± 0.1 ab
Chewiness	408.9 ± 147.9 a	158.6 ± 136.3 b	236.0 ± 133.9 b	178.1 ± 99.0 b
Gumminess	533.9 ± 172.3 a	277.0 ± 150.0 b	356.7 ± 162.7 ab	350.4 ± 127.3 ab

* The results are the mean of ten determinations ± the standard deviation. The comparison between the different storage times was carried out by one-way ANOVA. Different lowercase letters indicate a statistically significant difference (*p* < 0.05). Fresh = fresh tortellini (not frozen); Frozen T0 = frozen tortellini at the time of freezing treatment; Frozen T1 = frozen tortellini after 35 days of storage; Frozen T2 = frozen tortellini after 70 days of storage.

**Table 2 foods-12-01087-t002:** Evolution of phenolic compounds and carotenoids (mg/kg) in the soup *.

	Fresh	Frozen		
		T0	T1	T2
β-carotene	274.9 ± 5.3 a	270.7 ± 3.6 a	267.6 ± 4.7 a	266.3 ± 7.0 a
Lutein	41.5 ± 0.6 a	42.0 ± 1.3 a	40.3 ± 2.5 a	39.7 ± 1.6 a
α-Tocopherol	12.8 ± 0.1 a	12.9 ± 0.3 a	12.6 ± 0.5 a	12.4 ± 0.6 a
Quercetin-3-O-rutinoside	50.7 ± 1.4 a	52.5 ± 3.9 a	51.2 ± 1.8 a	49.9 ± 2.4 a
Quercetin	11.3 ± 0.8 a	10.4 ± 1.5 a	10.5 ± 0.4	10.0 ± 1.3 a

* The results are the mean of two determinations ± the standard deviation. The comparison between the different storage times was carried out by one-way ANOVA. Different lowercase letters indicate a statistically significant difference (*p* < 0.05). Fresh = fresh soup (not frozen); Frozen T0 = frozen soup at the time of freezing treatment; Frozen T1 = frozen soup after 35 days of storage; Frozen T2 = frozen soup after 70 days of storage.

**Table 3 foods-12-01087-t003:** Evolution of volatile compounds (µg/kg expressed as 4-methyl-2-pentanol) of the soup *.

	Fresh	Frozen		
		T0	T1	T2
*Aldehydes*				
Acetaldehyde	40.4 ± 0.6 a	44.4 ± 3.4 a	42.9 ± 2.2 a	44.1 ± 2.2 a
Pentanal	229.0 ± 3.7 a	235.6 ± 6.6 a	230.7 ± 9.1 a	230.0 ± 12.7 a
Hexanal	200.0 ± 18.0 a	192.0 ± 6.5 a	206.2 ± 11.0 a	222.0 ± 12.7 a
(E)-2-Heptenal	36.7 ± 3.2 a	32.9 ± 2.4 a	33.4 ± 1.8 a	36.3 ± 2.9 a
Nonanal	87.2 ± 6.3 a	87.6 ± 4.0 a	90.3 ± 6.8 a	89.5 ± 1.4 a
2,4-Decadienal	56.2 ± 3.4 a	61.1 ± 2.1 a	64.6 ± 1.0 a	68.4 ± 5.1 a
*Sum of aldehydes*	649.5 ± 19.9 a	653.6 ± 11.1 a	668.1 ± 16.1 a	690.2 ± 19.1 a
*Alcohols*				
2-Methyl-1-butanol	8.0 ± 0.8 a	8.0 ± 0.3 a	8.2 ± 0.5 a	7.0 ± 0.7 a
1-Pentanol	116.0 ± 5.2 a	108.3 ± 4.3 a	117.1 ± 15.5 a	126.9 ± 2.1 a
1-Hexanol	142.6 ± 3.0 ab	159.5 ± 13.1 a	138.2 ± 3.6 ab	119.9 ± 1.2 b
*Sum of alcohols*	266.5 ± 6.1 a	275.8 ± 13.8 a	263.5 ± 16.0 a	253.8 ± 2.5 a
*Ketones*				
Acetoin	106.6 ± 4.4 a	111.8 ± 6.4 a	123.5 ± 18.6 a	101.6 ± 9.0 a
*Terpenes*				
Limonene	258.5 ± 3.3 a	254.2 ± 9.4 a	310.7 ± 15.9 b	257.8 ± 17.5 a
α-Pinene	6424.7 ± 157.7 a	6413.0 ± 128.5 a	6439.0 ± 106.9 a	6449.5 ± 117.6 a
β-Pinene	1436.1 ± 50.2 a	1454.5 ± 46.5 a	1396.4 ± 121.3 a	1408.9 ± 237.9 a
β-Myrcene	3302.7 ± 33.3 a	3326.1 ± 54.4 a	3429.1 ± 8.7 a	3300.5 ± 46.2 a
γ-Terpinene	1243.7 ± 14.2 a	1228.3 ± 96.6 a	1227.3 ± 46.9 a	1205.0 ± 77.8 a
Caryophyllene	652.0 ± 11.4 a	667.1 ± 2.5 a	646.5 ± 43.7 a	674.1 ± 23.0 a
*Sum of terpenes*	13,317.7 ± 169.8 a	13,343.2 ± 176.3 a	13,449.1 ± 174.9 a	13,295.9 ± 281.9 a
*Sulphur compounds*				
Dimethyl sulfide	97.4 ± 3.4 a	102.9 ± 6.7 a	99.9 ± 2.5 a	110.1 ± 7.4 a
Dipropyl disulfide	40.9 ± 2.5 a	43.1 ± 3.0 a	41.2 ± 2.2 a	50.7 ± 3.9 a
*Sum of sulphur compounds*	138.3 ± 4.2 a	146.0 ± 7.4 a	141.0 ± 3.3 a	160.8 ± 8.3 a
*Furans*				
Furfural	69.5 ± 4.5 a	70.6 ± 4.0 a	66.9 ± 0.0 a	75.8 ± 7.6 a
2-Pentyl-furan	304.4 ± 18.8 a	299.0 ± 12.8 a	399.3 ± 24.4 b	360.5 ± 28.6 ab
*Sum of the furans*	373.9 ± 20.3 a	369.6 ± 17.0 a	466.1 ± 24.8 a	436.3 ± 31.9 a

* The results are the mean of two determinations ± the standard deviation. The comparison between the different storage times was carried out by one-way ANOVA. Different lowercase letters indicate a statistically significant difference (*p* < 0.05). Fresh = fresh soup (not frozen); Frozen T0 = frozen soup at the time of freezing treatment; Frozen T1 = frozen soup after 35 days of storage; Frozen T2 = frozen soup after 70 days of storage.

**Table 4 foods-12-01087-t004:** Evolution of volatile compounds (µg/kg expressed as 4-methyl-2-pentanol) of the tortellini *.

	Fresh		Frozen	
		T0	T1	T2
*Aldehydes*				
Acetaldehyde	34.3 ± 3.3 a	32.2 ± 3.3 a	29.2 ± 2.3 a	30.6 ± 3.4 a
Pentanal	72.1 ± 6.4 a	69.5 ± 6.7 a	77.3 ± 0.8 a	80.7 ± 7.5 a
Hexanal	209.2 ± 4.1 a	198.8 ± 10.8 a	219.7 ± 4.4 a	230.6 ± 10.3 a
(E)-2-Heptenal	3.3 ± 0.3 a	3.9 ± 0.3 a	4.0 ± 0.7 a	4.7 ± 0.1 a
Nonanal	34.9 ± 0.2 a	36.4 ± 4.2 a	38.4 ± 0.5 a	40.5 ± 3.7 a
2,4-Decadienal	2.8 ± 0.3 a	3.2 ± 0.3 a	3.4 ± 0.1 a	3.6 ± 0.2 a
*Sum of aldehydes*	356.6 ± 8.3 ab	344.0 ± 13.7 b	372.1 ± 5.1 ab	390.6 ± 13.7 a
*Alcohols*				
1-Pentanol	45.7 ± 2.0 a	47.9 ± 2.2 a	43.5 ± 2.9 a	40.6 ± 2.9 a
1-Hexanol	147.1 ± 6.3 a	153.0 ± 2.4 a	155.2 ± 6.4 a	142.1 ± 1.8 a
*Sum of alcohols*	192.8 ± 6.6 a	200.9 ± 3.3 a	198.7 ± 7.0 a	182.7 ± 3.4 a
*Ketones*				
Acetoin	159.5 ± 15.9 a	171.7 ± 16.3 a	186.9 ± 15.5 a	184.7 ± 7.8 a
2-Pentanone	474.8 ± 41.1 a	482.4 ± 30.2 a	474.4 ± 20.0 a	486.0 ± 22.5 a
2-Heptanone	896.1 ± 34.9 a	864.9 ± 22.0 a	898.1 ± 7.3 a	900.0 ± 19.3 a
2-Nonanone	241.6 ± 7.5 a	251.7 ± 11.7 a	262.6 ± 20.6 a	260.4 ± 12.5 a
*Sum of ketones*	1772.0 ± 56.7 a	1770.6 ± 42.4 a	1822.0 ± 33.4 a	1831.1 ± 33.1 a
*Terpenes*				
Limonene	2515.1 ± 106.0 a	2520.9 ± 29.7 a	2509.5 ± 50.3 a	2560.1 ± 53.0 a
α-Pinene	14,445.0 ± 492.5 a	14,418.2 ± 555.6 a	14,428.0 ± 354.5 a	14,537.1 ± 445.6 a
β-Pinene	2985.8 ± 77.5 a	2956.1 ± 91.3 ab	2870.2 ± 17.5 ab	2836.7 ± 12.1 b
Camphene	291.1 ± 15.5 a	303.8 ± 17.6 a	309.7 ± 20.9 a	299.5 ± 22.7 a
β-Myrcene	1357.7 ± 30.0 a	1290.5 ± 6.5 a	1292.3 ± 85.4 a	1255.1 ± 61.4 a
α-Phellandrene	699.8 ± 3.8 a	705.8 ± 9.7 a	722.2 ± 66.2 a	729.0 ± 33.9 a
β-Phellandrene	3093.9 ± 175.0 a	2978.9 ± 61.4 a	3077.7 ± 67.6 a	3104.9 ± 38.6 a
γ-Terpinene	3467.4 ± 231.0 a	3451.8 ± 65.1 a	3360.6 ± 174.2 a	3253.4 ± 132.9 a
Linalool	240.1 ± 8.7 a	244.1 ± 7.5 a	236.0 ± 11.2 a	222.0 ± 14.5 a
*Sum of terpenes*	29,095.9 ± 587.4 a	28,870.0 ± 571.3 a	28,806.2 ± 419.1 a	28,797.8 ± 475.7 a
*Organic acids*				
Acetic acid	46.7 ± 3.9 a	40.9 ± 2.6 a	47.7 ± 1.4 a	45.1 ± 3.4 a
Butanoic acid	41.4 ± 3.7 a	36.7 ± 2.9 a	43.8 ± 0.4 a	42.3 ± 4.7 a
Pentanoic acid	50.5 ± 3.2 a	47.9 ± 4.1 a	50.6 ± 3.3 a	55.7 ± 4.2 a
*Sum of organic acids*	138.6 ± 6.3 a	125.5 ± 5.7 a	142.2 ± 3.6 a	143.2 ± 7.2 a

* The results are the mean of two determinations ± the standard deviation. The comparison between the different storage times was carried out by one-way ANOVA. Different lowercase letters indicate a statistically significant difference (*p* < 0.05). Fresh = fresh tortellini (not frozen); Frozen T0 = frozen tortellini at the time of freezing treatment; Frozen T1 = frozen tortellini after 35 days of storage; Frozen T2 = frozen tortellini after 70 days of storage.

**Table 5 foods-12-01087-t005:** Results of the triangle test performed on frozen soup and tortellini samples at the time of freezing treatment (FS T0 and FT T0) compared to fresh ones (NFS and NFT), and the maximum number of correct responses required (values extracted from ISO 4120:2021).

Products		Responses Total Number	Correct Responses	Maximum Number of Responses	*α* = 0.01
Soup	Fresh vs. Frozen	25	22	17	significant
Tortellini	Fresh vs. Frozen	25	22	17	significant

## Data Availability

No new data were created or analyzed in this study. Data sharing is not applicable to this article.

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
