# Peer review of "Frozen Ready-to-(h)eat Meals: Evolution of Their Quality during a Real-Time Short Shelf Life"

_foods, 2023, doi:10.3390/foods12051087_

Round 1

Reviewer 1 Report

The work is to investigate the quality, flavor and sensory evolution of frozen dumplings and soup during the 70-day shelf. The results can provide some useful information. However, the innovation of this paper is low. The principle and mechanism of product quality change have not been discussed in depth.  For author's reference, some comments are provided below:

Abstract

- Lines 15-17, abstract writing should be refined, not popular science and general description. The author only needs to define the specific works that have been carried out and the main results obtained.

- Lines 17-25, the experimental results in the summary are not clear, the description is more general, and lack of intuitive data and mechanism support.

Introduction

- The introduction is long, please shorten it. The introduction should closely focus on the experimental background and research purpose, and the author should some of the irrelevant parts.

- Lines 63-93, the goal of the manuscript is to observe changes in the quality of frozen food during storage. Therefore, the redundant introduction of the composition of microwave technology, the principle of microwave heating and the application of packaging technology should be deleted.

Materials and Methods

- Lines 159-160, the author needs to supplement the manufacturer of the Bostwick con-sistometer instrument. Also, it is necessary to check the omission of the instrument information of the full text of the manuscript.

Results and discussion

- Lines 289-291, the experimental data have such a large standard deviation, how to ensure the repeatability of the results?

- Lines 294-296, not all tortellini received the same amount of heat and did not heat up in the same way. Different conditions will lead to different quality of the product, so how to determine the scientific nature of obtaining experimental data?

- Lines 296-298, differences in the amount of filling meat leads to the experimental variable is not single, how to ensure the reliability of texture data?

- Lines 361-372, Table 3 and Table 4, please explain why acidity and peroxide value decrease first and then increase at T0 (Table 3). Why is there a significant difference in peroxide value in Table 3 soup but not in Table 4 tortellini? The mechanism needs to be analyzed and supplemented.

- Lines 456-459, 468-484, 492-495, there is no significant difference in volatile components between soup and dumpling during storage. Does it mean that the freezing conditions in this experiment can also effectively reduce the divergence of volatile substances in other frozen foods?

Conclusions

- Lines 588-596, freezing temperature and freezing time will affect the quality and sensory properties of the product. The author only briefly examined the normal refrigerator freezing temperature (- 18 ℃) and short-term time changes (35-70 days). More positive controls with different temperatures and different duration should be analyzed, and the experimental results may be more obvious, so that they can be more convincing.

Author Response

REPLIES TO REVIEWER 1

Abstract

- Lines 15-17, abstract writing should be refined, not popular science and general description. The author only needs to define the specific works that have been carried out and the main results obtained.

- Lines 17-25, the experimental results in the summary are not clear, the description is more general, and lack of intuitive data and mechanism support.

Thank you for your comment. According to your suggestion, we rewrite the abstract. We do hope that this version is clearer than the previous one.

Introduction

- The introduction is long, please shorten it. The introduction should closely focus on the experimental background and research purpose, and the author should some of the irrelevant parts.

Thank you for your observation. Cuts have been made to reduce the length of the introduction.

- Lines 63-93, the goal of the manuscript is to observe changes in the quality of frozen food during storage. Therefore, the redundant introduction of the composition of microwave technology, the principle of microwave heating and the application of packaging technology should be deleted.

Thank you for your comment. The parts you suggested have been deleted.

Materials and Methods

- Lines 159-160, the author needs to supplement the manufacturer of the Bostwick con-sistometer instrument. Also, it is necessary to check the omission of the instrument information of the full text of the manuscript.

Thank you for your comment. This part has been inserted (lines 138-139).

Results and discussion

- Lines 289-291, the experimental data have such a large standard deviation, how to ensure the repeatability of the results?

Thank you for your observation. As mentioned in the paper, the high standard deviation depends on some intrinsic and extrinsic characteristics of the product. The repeatability is ensured by the fact that the results obtained are the average of 10 different repeats made on each sample.

- Lines 294-296, not all tortellini received the same amount of heat and did not heat up in the same way. Different conditions will lead to different quality of the product, so how to determine the scientific nature of obtaining experimental data?

Thank you for your observation. It is assumed that tortellini did not receive the same heat amount because of the shape of the container used as packaging but not for the microwave method which usually allows a rapid and a homogeneous heating. However, we could consider the scientific nature of the data assured, because of the 10 repetitions of the analysis on tortellini made on several parts of the product.

- Lines 296-298, differences in the amount of filling meat leads to the experimental variable is not single, how to ensure the reliability of texture data?

Thank you for your observation. When we talk about differences related to the amount of filling in meat, we mean differences related to the distribution in terms of space of the meat inside the tortellino, which can vary, even if the quantities are the same and if the filling process is carried out mechanically.

- Lines 361-372, Table 3 and Table 4, please explain why acidity and peroxide value decrease first and then increase at T0 (Table 3). Why is there a significant difference in peroxide value in Table 3 soup but not in Table 4 tortellini? The mechanism needs to be analyzed and supplemented.

Thank you for your comment. The explanation was given in lines 366-373.

- Lines 456-459, 468-484, 492-495, there is no significant difference in volatile components between soup and dumpling during storage. Does it mean that the freezing conditions in this experiment can also effectively reduce the divergence of volatile substances in other frozen foods?

Thank you for your comment. Regarding the volatile compounds of the two products, we see that there are no significant differences between fresh and frozen meals and during the storage period. It is possible to assume, in accordance with the results obtained, that the freezing conditions applied in this experimentation, can lead to the same results in other types of products.

Conclusions- Lines 588-596, freezing temperature and freezing time will affect the quality and sensory properties of the product. The author only briefly examined the normal refrigerator freezing temperature (- 18 ℃) and short-term time changes (35-70 days). More positive controls with different temperatures and different duration should be analyzed, and the experimental results may be more obvious, so that they can be more convincing.

Thank you for your comment. According to our opinion, it was not necessary to carry out further trials at different storage temperatures, since all freezers in the household, in bars or restaurants are at - 18 °C/- 20 °C. It may be interesting to do further studies as the days of storage of the product increase.

Reviewer 2 Report

This study examined the quality of frozen ready meals during storage. It is suggested to modify. Some detailed comments are as follows:

1. Line 22 - include the p value.

2. Line 113-114 - Include all other ingredients used to produce the soup and tortellini.

3. Line 120 -  Is it necessary to include '+' symbol to the temperature?

4. Line 138-152 - Include the profile settings for the TPA.

5. Line 250 - State the venue of the sensory evaluation session.

6. Line 254 - Provide the composition of the judges involved for sensory evaluation.

7. Line 301 - Why the hardness of frozen samples decreased? Provide the details explanation. 

8. Superscript for all tables should be typed properly.

9 . Provide the description of samples coding for all tables.

10. Standardized the decimal points for all results presented in the tables.

11. Line 502 - Expert panelists of judges?

12. Line 511 - Should used trained panelists instead of experienced panelists.

Author Response

REPLIES TO REVIEWER 2

  1. Line 22 - include the p value.

Thank you for your comment. The value has been included.

  1. Line 113-114 - Include all other ingredients used to produce the soup and tortellini.

Thank you for your comment. Other ingredients were added (lines 84-91).

  1. Line 120 -  Is it necessary to include '+' symbol to the temperature?

Thank you for your comment. Symbol has been deleted.

  1. Line 138-152 - Include the profile settings for the TPA.

Thank you for your comment. Settings are included (lines 113-118).

  1. Line 250 - State the venue of the sensory evaluation session.

Thank you for your comment. The venue has been included (lines 230-231).

  1. Line 254 - Provide the composition of the judges involved for sensory evaluation.

Thank you for your comment. The composition of the judges involved has been included (lines 242-243 and lines 250-251).

  1. Line 301 - Why the hardness of frozen samples decreased? Provide the details explanation. 

Thank you for your comment. Attempts have been made to give a more detailed explanation of lines 285-288.

  1. Superscript for all tables should be typed properly.

Thank you for your comment. Subscripts of all tables have been typed properly.

  1. Provide the description of samples coding for all tables.

Thank you for your comment. A description of the coding samples for all tables has been provided.

  1. Standardized the decimal points for all results presented in the tables.

Thank you for your comment. Decimal points have been standardized for all the results presented in the tables.

  1. Line 502 - Expert panelists of judges?
  2. Line 511 - Should used trained panelists instead of experienced panelists.

Thank you for your comment. The corrections suggested have been made.

Reviewer 3 Report

1. The aim of the study is insufficient. In this study, it is seen that these two products compared are qualitatively different. Why are these two products, which are different in quality, compared? Results could have been given more clearly in the abstract.

2. The abstract should be revised and rewritten in a more understandable way.

3. The introduction can be shortened and the purpose should be clearly written.

4. Line 110: I think the number of samples is insufficient. The purpose of determining this number of samples should be written.

5. Discussions are inadequate and not clear enough. Evaluation should be done in conjunction with current studies.

Revisions requested to be made in the article were shown in the manuscript. The manuscript can be revised.

Author Response

REPLIES TO REVIEWER 3

  1. The aim of the study is insufficient. In this study, it is seen that these two products compared are qualitatively different. Why are these two products, which are different in quality, compared? Results could have been given more clearly in the abstract.

Thank you for your observation. Two very different products were considered to observe how the freezing process can have a different influence on the quality of the product according to the different types of ingredients contained.

  1. The abstract should be revised and rewritten in a more understandable way.

Thank you for your observation. The abstract has been rewritten more clearly.

  1. The introduction can be shortened and the purpose should be clearly written.

Thank you for your observation. The introduction has been deeply modified, according to your suggestion.

  1. Line 110: I think the number of samples is insufficient. The purpose of determining this number of samples should be written.

Thank you for your comment. The paragraph 2.1 has been rewritten.

  1. Discussions are inadequate and not clear enough. Evaluation should be done in conjunction with current studies.

Thank you for your comment. To the best of our knowledge, our manuscript  reported, for the first time, a complete profile of  quality evolution of the two ready to (h)eat meals considered, thus for some experimental evaluations, we couldn’t find proper already published data to cite to validate or rebut.

Reviewer 4 Report

The study explored the evolution of the quality of tortellini and carrot soup during 70 days shelf life test. It supplements the understanding of the quality and nutritional changes of ready-to-eat food. A series of indicators were measured, such as consistency texture and acidity the peroxide value of the oilphenols and carotenoids and volatile compounds, etc. The paper needs to be carefully revised before further consideration for publication. The discussion of some results is not deep enough, and it is suggested to modify. Some detailed comments are as follows:

1.     In section 2.2, for the selection of products to be determined, they are divided into T0, T1, T2, where T0 used the fresh samples and a part of the freshly frozen samples , It mean to take a mixture? However, in subsequent determination, fresh and unfrozen samples are used as the blank group for comparison. Why is T0 not simply selected from unstored but frozen samples?

2.     In section 3.1, evolution of the texture of tortellini were tested, Table 1 shows that there are significant differences between the indexes of T0 samples and fresh samples. With time, the indexes of T0, T1 and T2 have some significant changes, but some have no changes. The discussion in this part should be strengthened.

3.     In the full text, the data is mostly in the form of tables, and some tables are suggested to be summarized in other forms. For example, Table 2, Table 3, etc. can be represented by graphs to show the results more accurately and clearly.

4.     In Table 5, there is no significant difference between the extracted ingredients in several groups of samples, which indicates that the product is relatively stable in terms of composition, and can also show that the nutrition of ready-to-eat food is not affected by the long shelf life. The discussion can be inclined to this aspect.

5.     In Figure 3 and Figure 4, among all indicators, the indicators with significant differences need to be highlighted, which can be more easily distinguished.

Author Response

REPLIES TO REVIEWR 4

  1. in section 2.2, for the selection of products to be determined, they are divided into T0, T1, T2, where T0used the fresh samples and a part of the freshly frozen samples, It mean to take a mixture? However, in subsequent determination, fresh and unfrozen samples are used as the blank group for comparison. Why is T0 not simply selected from unstored but frozen samples?

Thank you for your comment. This part has been rewritten more clearly (lines 99-103).

  1. In section 3.1,evolution of the texture of tortellini were tested, Table 1 shows that there are significant differences between the indexes of T0 samples and fresh samples. With time, the indexes of T0, T1 and T2 have some significant changes, but some have no changes. The discussion in this part should be strengthened.

Thank you for your comment. We tried to strengthen the discussion in lines 285-293, however data relative to the texture of tortellini were original and no similar studies were found on the same type of product.

  1. In the full text, the data is mostly in the form of tables, and some tables are suggested to be summarized in other forms.For example, Table 2, Table 3, etc. can be represented by graphs to show the results more accurately and clearly.

Thank you for your observation. Table 2, Table 3 and Table 4 were changed in figures.

  1. In Table 5, there is no significant difference between the extracted ingredients in several groups of samples, which indicates that the product is relatively stable in terms of composition and can also show that the nutrition of ready-to-eat food is not affected by the long shelf life. The discussion can be inclined to this aspect.

Thank you for your comment. According to the data relative to the phenolic compounds and carotenoids, we could conclude that the freezing process adopted and subsequent storage for 70 days at - 18 °C did not have a negative impact on the nutritional and health quality of the soup. Thus, it is plausible to suppose that the freezing process did not change the macronutrients’ composition. However, further proper studies could be carried out.

  1. In Figure 3 and Figure 4,among all indicators, the indicators with significant differences need to be highlighted, which can be more easily distinguished.

Thank you for your comment. The indicators have been circled in the figures.

Round 2

Reviewer 1 Report

Accept in present form.

Reviewer 3 Report

This manuscript can be accepted.

Reviewer 4 Report

Accept in present form